# Evaluation of sustainability of mycorrhizal industry development in Fujian Province-based on a combination of CRITIC empowerment method and cloud model

Xue-Yuan Li[1,2]*, Sen-wei Huang[1,2], Zu-Mei Cai[1,2], Yue-Wei Hong[1,2], Qian Lin[3]

1 School of Public Administration and Law, Fujian Agriculture and Forestry University, Fuzhou, China,
2 Rural Development Research Center, Fujian Agriculture and Forestry University, Fuzhou, China, 3 School of Foreign Languages, Huaqiao University, Quanzhou, China

* 764941740@qq.com

## Abstract

The sustainable development of mycorrhizal industry is the key to solve the problem of "mycorrhizal forestry contradiction". As a major province of edible mushroom production and forestry resources in China, Fujian Province is also an important origin of mycorrhizal technology research and development, so it is more typical and practical to establish an index system to evaluate the sustainability of mycorrhizal industry development in Fujian Province. Through research interviews and data collection, a sustainable capacity evaluation system of mycorrhizal industry was established with 21 indicators in six dimensions: economic, ecological, social, cultural, political, and technological. A combination of CRITIC empowerment method and cloud model was used to evaluate the sustainability of mycorrhizal industry development in Fujian Province. The results show that although the economic sustainability of the mycorrhizal industry in Fujian Province is average, the overall development trend is good and there are not too many problems. The sustainability of ecological, social and technological levels all have large differences in the development of indicators and the overall development status is average, but overall, the ecological, social and technological levels show a steady forward development from 2017 to 2020. The cultural and political dimensions of sustainability not only have large differences in the development of indicators and an average overall development status, but also have a small development span from 2017 to 2020 and a slow overall development.

## Introduction

In the 1980s, China introduced cultivation technology from abroad, and farmers consumed a lot of wood to produce edible mushrooms such as shiitake and fungus in order to get rid of poverty and get rich. As a large forestry province, Fujian Province has been in the forefront of edible mushroom production in China, which has led to the problem of "contradiction between mushroom and forest" in Fujian Province. It was not until 1986 that Jim Lin invented

**Data Availability Statement:** The data on which the results presented in the study are based can be

obtained from statistical yearbooks and official government websites.

**Funding:** The article is funded by the National Social Science Foundation of China General Project "Research on Modernization of Rural Environmental Governance System and Governance Capacity" (20BSH113)The project leader of this fund project is Huang Senwei, who has made the following contributions in writing the article: making preliminary modifications to the article and grasping the overall direction of the article.

**Competing interests:** The authors have declared that no competing interests exist.

the mycorrhizal technology, which can use grass instead of wood to raise mushrooms. The advantage of mycorrhizal over wood is that it has a short growth cycle and low requirements for external conditions. In this regard, the birth of mycorrhiza has theoretically successfully solved the increasingly prominent problem of "mycorrhizal contradiction" [1]. However, although mycorrhiza is rich in nutrients, it has both economic output value and ecological service value as well as social carrying value. However, in the current situation of Fujian Province, the demand for medicinal mushrooms is huge, but the market chooses to continue to cut down the forest wood in order to maintain the supply of medicinal mushrooms, and the mycorrhizal grasses are rarely used as culture media for medicinal mushrooms. Due to the irreversibility of wood cutting, the failure to improve the replacement rate of mycorrhiza for wood will only aggravate the potential problem of soil erosion in counties and cities, and the "mycorrhiza contradiction" in Fujian Province is increasing [2]. In this regard, how to promote the sustainable development of mycorrhizal industry and improve the replacement rate of mycorrhizal for wood plays a decisive role in solving the "mycorrhizal contradiction" problem. The worsening of the "mycorrhizal contradiction" problem also indicates that the relationship between economy and environment has not been well coordinated in the process of economic and social development in Fujian Province. The sustainable development of mycorrhizal industry is not only related to the development of agriculture and forestry economy, but also related to the balance of ecological environment, and more importantly, to the development of diplomatic work. In this regard, constructing an index system to evaluate the sustainability of mycorrhizal industry can effectively analyze the current development of mycorrhizal industry, but there are not many studies on mycorrhizal and mycorrhizal industry, and there is still no research to build a set of evaluation system to evaluate the sustainable development of mycorrhizal industry. The study hopes to enrich the research scope of mycorrhizal industry by selecting indicators based on the functional role of mycorrhiza itself, and expects to suggest suggestions to improve the sustainable development of mycorrhizal industry. The combination of CRITIC empowerment method and cloud model is able to objectively empower the data to achieve scientific and reliable evaluation results, and use the cloud model to analyze and interpret the results at a deeper level, which is a realistic guidance for analyzing the reasons of the slow development of mycorrhizal industry and promoting the sustainable development of mycorrhizal industry in the future.

## Literature review and analytical framework

Through a review of the existing literature on the development of the mycorrhizal industry, the research perspective of the mycorrhizal industry has gradually expanded from agricultural economy and ecology to social, political and cultural fields. Scholars believe that the role of mycorrhiza should be considered from the perspective of its multi-functionality, which can improve the laws and regulations related to mycorrhiza cultivation and promote mycorrhiza cultivation technology [3]. The application of mycorrhiza in environmental protection and soil and water conservation is presented in [4] and the ecological value of mycorrhiza in arid areas and sandy areas is evaluated in [5], especially the ecological value of mycorrhiza in preventing soil erosion and salinization in the Yellow River Basin [6]. It is further suggested that the promotion of mycorrhizal industry in agricultural economy needs to be combined with the livelihood capital capacity of micro-organism farmers, such as risk tolerance, to enhance the recognition of mycorrhizal, and to increase agricultural education and training to improve comprehensive literacy to promote the sustainable development of mycorrhizal industry [7]. Some scholars also introduced the value and significance of developing mycorrhizal industry in four aspects: economic, cultural and leisure, social and ecological through the profound

connotation of mycorrhizal industry to agricultural green development [1]. Meanwhile, the mycorrhizal industry has played an important political role in China's diplomatic cooperation with developing countries, and the contribution of the mycorrhizal industry has been recognized internationally [8, 9]. Besides, mycorrhiza, which has an important ecological role, has been further valued by scholars for its value in edible mushroom food culture and tourism [1, 10]. As early as 1996, Lin Changxin scholars linked the mycorrhizal industry and the view of sustainable development and proposed six principles about the sustainable development of mycorrhizal industry as a theoretical basis for the study of mycorrhizal sustainable development [11]. Mycorrhizal industry is the upgrade of traditional mycorrhizal industry and the innovation development of traditional mycorrhizal industry, how to develop mycorrhizal industry sustainably is also the topic of heated discussion in academic circles [12]. In recent years, scholars have used fuzzy hierarchical analysis to construct the index system of sustainable development in the field of mycorrhizal industry for the first time, and elaborated the sustainability of mycorrhizal agricultural technology in foreign aid [13]. Combining the above-mentioned scholars' research results and the reference of evaluation system and considering the functionality of mycorrhiza itself [2], this paper establishes a system of 21 indicators for sustainable development of mycorrhiza industry in six dimensions, including technical, social, economic, ecological, political and cultural (see Table 1).

The current academic research on multi-criteria decision making often uses the assignment method to divide the weights of multiple indicator systems in order to achieve the scientific nature of the evaluation results. The mainstream view is that the assignment method can be divided into two types: subjective assignment method and objective assignment method, and subjective assignment method, such as AHP method [14], FUCOM method [15], Delphi method, etc., which mainly relies on the information provided by experts to assign weights to indicators, and its shortcomings are obvious. This means that the decision maker influences the decision making process and the outcome is more influenced by the decision maker's subjective preferences. The objective empowerment method uses existing data to make the empowerment. In comparison, the advantage of the objective assignment method over the subjective assignment method is that it is not influenced by the decision maker's preferences and is more scientific and reliable in the representation of the evaluation results [16]. Although scholars have pointed out that the objective assignment methods such as entropy method and CRITIC assignment method have the same defects in practice, CRITIC assignment method is more systematic than entropy method because it not only considers the comparison intensity between indicators but also considers the conflict between indicators, and has the advantages of both entropy method and principal component analysis. The use of CRITIC assignment method can The use of CRITIC weighting method can reflect the evaluation results more accurately and objectively and scientifically [17]. The advantage of the cloud model is that it can simulate the overall results through multiple simulations, reflect the stability and scientific nature of the overall results, and can realize the mutual transformation of quantitative and qualitative through its own data. Compared with the common line graphs and other ways to interpret the evaluation results, the evaluation of the cloud model can be more scientific and reliable, and it can objectively analyze the good and bad evaluation results and visualize the results, so that it is easy to analyze and dig into the overall situation of the development of the evaluation results, the overall stability, the differences in development, the stability of the development differences, and many other information. In the current research, the empowerment method is often combined with the cloud model, and the common combination is mainly the entropy method and cloud model, and the combination of CRITIC empowerment method and cloud model. Based on the introduction of the empowerment method above and considering the main combination of scholars at present, the study adopts the combination

**Table 1. Sustainability evaluation index system of mycorrhiza industry in 2017–2020.**

| Target layer | Guideline layer | Indicator layer | Specific content | Tertiary indicators | 2017 | 2018 | 2019 | 2020 | Unit | Calculation method | Data source |
|---|---|---|---|---|---|---|---|---|---|---|---|
| Evaluation of the sustainability of mycorrhiza industry development | Economic dimension sustainable | Food supply sustainability and stability | Supply of medicinal mushroom products and mushroom feed | Mycorrhiza production * Price change index | 123.16 | 126.28 | 133.36 | 137.88 | / | Edible mushroom prices | http://www.mushroommarket.net/average/list.php. |
| | | Sustainable and stable supply of raw materials | Supply of raw materials for food processing industry, pharmaceutical and chemical industry, biomass energy | | | | | | | | |
| | | Cost-benefit ratio of mycorrhiza cultivation | Increasing income from mycorrhiza cultivation | Planting benefits / planting costs | 6.1 | 6.15 | 6.2 | 6.25 | / | Planting benefit in 2020 divided by planting cost in 2020 divided by price change growth rate | https://baijiahao.baidu.com/s?id=1732131134728345750&wfr=spider&for=pc |
| | | Economic Situation | The performance and direction of the regional economy | Agricultural GDP growth rate | 38.7 | 39.1 | 38.3 | 37.1 | Billion | Share of agricultural GDP | Fujian Provincial Statistical Yearbook, EPS Database |
| | | | | Forestry GDP growth rate | 8.3 | 9.2 | 9 | 8 | Billion | Forestry GDP share | Fujian Provincial Statistical Yearbook, EPS Database |
| | Ecological dimension sustainable | Resource saving capacity | The ability to protect forest resources and replace wood resources | Mycorrhizal forest replacement rate of 30% | 28 | 28.53 | 29.586 | 30 | % | Edible mushroom medium mycorrhizal replacement rate, using the 2020 mycorrhizal forest replacement rate of 30% divided by thesis growth rate to obtain | ZHONG Hua, LI Li, QIN Xue, TAO Wenguang. A preliminary investigation of mushroom cultivation technology with grass instead of wood[J]. Farming and Cultivation,2020,40(05):19~22+26. |
| | | Environmental Protection Capability | Prevent soil erosion, purify the atmosphere and water sources, and maintain ecological balance | Carbon sequestration capacity, nitrogen fixation capacity, erosion control area | 3651.7 | 3778.4 | 3895.9 | 4031.8 | Thousands of hectares | Soil erosion control area | EPSS Data Platform |
| | | Natural Resource Endowment | Comparative advantages in natural resources compared to other regions | Available land area | 926.82 | 924.4 | 926.215 | 926.063 | million hectares | Forestry area 2019 and 2020 are taken as the average of the first four years | EPSS data platform agroforestry database forest land area |
| | Social Dimension Sustainability | Employment poverty reduction capacity | Absorbing surplus labor, alleviating poverty, and promoting social stability | Number of employed persons increased | 188.9951802 | 168.5129629 | 183.3088289 | 158.27 | 10,000 people | Employment in primary industry x proportion of forest economy output value to primary industry economic output value | Fujian Provincial Statistical Yearbook, EPS Database |
| | | Infrastructure | Hardware facilities supporting industrial development, mainly including transportation, water, electricity, Internet and other aspects | Road network density | 108012 | 108901 | 109785 | 110118 | Kilometers | Highway mileage in Fujian Province | Fujian Provincial Statistical Yearbook |
| | | Public Services | Software facilities supporting industrial development, mainly including education, medical care, social security, pensions and other aspects | Social Security Funds | 3945581 | 4681506 | 5078865 | 5723365 | million yuan | Public Finance Expenditure on Social Security and Employment in Fujian Province | Fujian Provincial Statistical Yearbook |
| | | Nutritional health value | Improving dietary structure and overall health | Protein conversion rate | 9.33 | 9.51 | 9.86 | 10 | % | Mycorrhiza crude protein ratio divided by cumulative thesis growth rate in 2020 | https://www.sxzq1.com/b/80044.html |
| | Cultural dimension sustainable | Mycorrhiza brand culture building | Create mushroom culture and enhance the popularity of mushroom plants | 49 species of high quality mycorrhizal grass seeds | 0 | 43 | 45 | 49 | species | Quality mycorrhizal grass seeds in addition to the patent increase Aki number growth rate | http://mein.isenlin.cn/?uid=mein&aid=21E409AC78I45C6896B3F5064C362FC |
| | | Tourism Industry Construction | Construction of leisure tourism and ecotourism for carrying out the mycorrhizal industry | Number of Demonstration Areas | 35 | 39 | 44 | 49 | individual | The growth rate of the number of demonstration areas divided by the cumulative number of policies in 2020 | https://baijiahao.baidu.com/s?id=1710557661562317396&wfr=spider&for=pc |
| | | Customs and Culture | Local food culture, medicinal culture and customs, etc. | Edible mushroom production price index (previous year = 100) | 105.27 | 100.97 | 103.33 | 93.66 | / | Edible mushroom production price index | Fujian Provincial Statistical Yearbook |
| | Political dimension is sustainable | Financial Support | Financial support to the mycorrhiza industry | Financial support funds | 5158.23 | 4947.53 | 5063.17 | 4589.34 | million yuan | Public budget of agriculture, forestry, animal husbandry and fishery x The proportion of the output value of the forest economy to the output value of the primary economy | Fujian Provincial Statistical Yearbook, EPS Database |
| | | Endogenous power | The conscious will of the sovereign to develop | Learn the frequency of mycorrhizal mentions in strong countries | 0 | 4 | 2 | 3 | individual | Study strong official website search mycorrhizal statistics Xi Jinping log number | Learning Power App |
| | | Policy support efforts | Number of policies to support the development of mycorrhizal industry, publicity efforts and landing situation | Number of policies | 8 | 8 | 9 | 10 | individual | Policy Compendium Document (Cumulative) | The corresponding government departments and institutions official website collation |
| | | International promotion efforts | Promotion of mycorrhiza industry in the international arena | Number of international technical training courses on mycorrhizal technology | 153 | 174 | 202 | 245 | times | Number of training classes | Fujian Agriculture and Forestry University official website reports |
| | Technical level sustainable | Practitioner quality and competence | The overall education level and scientific research ability of mycorrhiza employees | Number of international technical trainers in mycorrhiza technology (persons) | 6602 | 6979 | 7817 | 8653 | People | Number of trainers | Learning Power App |
| | | Technology promotion and practical application | New technology promotion efforts and practical production applications | Increase of patents in mycorrhiza industry (pcs) | 34 | 13 | 9 | 7 | individual | Increase in the number of patents | https://cprs.patentstar.com.cn/Analysis/Index?CurrentQuery=6l+M6l2JL1lZ&type=cn |
| | | Technology Development Capability | The number of research institutions and colleges related to mycorrhiza industry, research level and research infrastructure construction | Number of papers (piece) | 54 | 107 | 162 | 217 | Part | Cumulative number of papers on the topic "Mycorrhiza" found by CNKI | China Knowledge Network |

model of CRITIC empowerment method and cloud model to analyze and evaluate the sustainability of mycorrhizal industry.

In summary, the contributions of this paper are: first, by summarizing the impact of scholars on six aspects of mycorrhizal industry development, including technological, social, economic, ecological, political, and cultural aspects, it fills the lack of research on the construction of sustainability indicators of mycorrhizal industry and enriches the research perspective and content of mycorrhizal topics. Second, objective time series statistical data are selected to evaluate the sustainability indicators of mycorrhiza, and a model combining CRITIC method and cloud model is used in the method to evaluate the innovation. Third, we propose corresponding policies and opinions on the sustainable development of mycorrhizal industry in Fujian Province based on the evaluation results of the cloud model, which is an important practical guidance to improve the sustainable development of mycorrhizal industry in Fujian Province.

The research data were selected from the relevant index data for 2016–2020, and the data were mainly obtained from the statistical yearbook of Fujian Province, ESP database, and some data were obtained from the relevant literature, relevant government part documents and official reports. The specific data situation, data sources and calculation methods are detailed in Table 1.

## Mycorrhizal sustainability analysis

### Assignment of evaluation indicators based on CRITIC method

There are two methods commonly used to assign weights to evaluation indicators: subjective weighting method and objective weighting method. In order to prevent the research from being too subjective and to ensure the objectivity and scientificity of the data results, objective weighting methods such as entropy weighting method, principal component analysis method and standard deviation method are often used to assign the evaluation indexes. The CRITIC method takes into account the correlation between data on the basis of the conflict and comparison intensity among evaluation indexes, and has the advantages of entropy weighting method and principal component analysis method, and is more systematic. Therefore, the CRITIC method was used to assign weights to the evaluation indexes in the paper [18].

According to the basic principle of CRITIC method combined with the sustainability evaluation characteristics of mycorrhiza industry, the specific assignment operation steps are as follows.

First, the initial index set matrix is constructed. According to the constructed evaluation index system survey to obtain m evaluation index data of n survey objects, the initial standard matrix R of data is constructed.

$$R = \begin{bmatrix} r_{11} & \cdots & r_{1m} \\ \vdots & \ddots & \vdots \\ r_{n1} & \cdots & r_{nm} \end{bmatrix} (i = 1, 2, \cdots, n; j = 1, 2, \cdots, m) \tag{1}$$

In Eq (1), rij indicates the evaluation of the jth indicator by the ith survey respondent; n is the number of survey respondents and m is the number of indicators.

Second, the comparison intensity of each indicator is calculated. The intensity of comparison between the jth indicator is calculated based on the data indicators, i.e., the standard

deviation of the data is calculated $\delta_j$.

$$\delta_j = \sqrt{1/n * \sum_{i=1}^{n} (r_{ij} - u)^2} \, (j = 1, 2, \cdots, m) \tag{2}$$

In Eq (2), u is the arithmetic mean of the indicators.

Third, the conflictiveness between indicators is calculated. Conflict between the jth indicator and other indicators in the evaluation system is calculated by the correlation coefficient between the evaluation data of indicators $y_j$.

$$y_j = \sum_{i=1}^{m} \left( 1 - \frac{Cov(j, t)}{\delta_j \delta_t} \right) \, (j = 1, 2, \cdots, m) \tag{3}$$

In Eq (3), Cov(j,t) is the covariance of data indicator j and data indicator t; t is the standard deviation of data indicator t.

Fourth, the calculation of integrated information quantity. Combine the contrast intensity and conflict of indicators to calculate the comprehensive information quantity of indicators $C_j$.

$$C_j = \delta_j y_j (j = 1, 2, \cdots, m) \tag{4}$$

Fifth, calculate the index weights. Calculate the weight of each indicator according to the comprehensive information of the indicator $W_j$.

$$W_j = \frac{C_j}{\sum_{i=1}^{m} C_j} \, (j = 1, 2, \cdots, m) \tag{5}$$

## Basic concepts and computation of cloud models

Cloud model is a common fuzzy mathematical algorithm analysis method, mostly used in the comprehensive evaluation of things, its main role is to achieve qualitative and quantitative mutual transformation, through the quantitative calculation of the results of the evaluation results in terms of the mean value ($E_x$), consistency ($E_n$) and the concentration of distribution ($H_e$) The three aspects of evaluation results are visualized in terms of mean, consistency and concentration of distribution, which facilitate the analysis and discussion [19, 20].

Two main methods exist for cloud modeling, the forward cloud generator method and the inverse cloud generator method. The inverse cloud generator is quantitative to qualitative diffraction, and the forward cloud generator is qualitative to quantitative diffraction. Among them, the inverse cloud generator is the process of reducing the three numerical features of C (Ex, En, He) by inputting a certain number of cloud droplets, and the specific calculation operation process of the inverse cloud generator is as follows.

First, calculate the cloud drop sample mean ($E_x$) with the sample variance ($S_n$)

Second, calculate the entropy of the cloud droplet sample ($E_n$)

$$E_n = \sqrt{\pi/2} \times \frac{1}{N} \sum_{i=1}^{N} x_i - E_x \tag{6}$$

In Eq (6), N is the total number of samples, and $x_i$ is the observed value of the ith sample.

Third, calculate the cloud droplet sample hyperentropy ($H_e$)

$$H_e = \sqrt{S_n^2 - E_n^2} \tag{7}$$

Fourth, the output cloud droplet digital features (Ex, En, He).

## Mycorrhizal industry sustainability evaluation method and process design

According to the cloud model principle, the whole mycorrhizal industry sustainability evaluation index system as a theoretical domain, each research object as a cloud droplet, the overall characteristics of the cloud formed by the integrated results of all research objects on all indicators evaluation is reflecting the mycorrhizal industry sustainability, according to which the mycorrhizal industry sustainability evaluation method process is designed as follows.

Step 1: Determine the set of factors

The evaluation of sustainability of mycorrhizal industry contains 6 types of primary indicators, i.e., the corresponding 21 secondary indicators. Because the research index system is different in terms of the measurement level, the data of the indexes need to be normalized, because the research indexes are all positive indicators, only one normalization process is needed, and there is no need to consider the normalization process of negative indicators, the normalization process is as follows.

$$Y = (X - X_{min})/(X_{max} - X_{min}) \tag{8}$$

Step 2: Determine the evaluation cloud and evaluation set

The set of established indicators $U = \{U_1, U_2, \cdots, U_m\}$ (m is the number of evaluation indicators), establish the corresponding evaluation set $V = \{V_1, V_2, \cdots, V_m\}$ corresponding to n different degrees of sustainability evaluation dimensions, according to the bilateral constraint criterion, the value of each rubric is within the limited family domain, and the minimum value of the rubric is set as $T_{min}$ and the maximum value is $T_{max}$. The three numerical characteristics of the evaluation criteria cloud are calculated as follows.

$$\begin{cases} E_x = (T_{min} + T_{max})/2 \\ E_n = (T_{max} - T_{min})/6 \\ \quad H_e = k \end{cases} \tag{9}$$

In Eq (8), k denotes randomness, and in this paper, k is taken as 0.01. The numerical characteristics of the cloud model corresponding to the evaluation level are shown in Table 2.

Step 3: Determine the set of weights

According to the table shown, the weights of the primary indicators are calculated using the CRITIC method, i.e. $w_1, w_2, \ldots, w_n$ Then the weights of the second-level indicators are calculated, i.e. $w_{11}, w_{12}, \ldots, w_{nm}$, and finally, the set of weight vectors is calculated by using the

**Table 2. Cloud model evaluation set.**

| Evaluation Levels | Assignment score interval | Evaluating the numerical characteristics of set cloud models |
|---|---|---|
| Sustainable and strong | (0.8, 1] | (0.9, 0.03, 0.01) |
| Stronger sustainable | (0.6, 0.8] | (0.7, 0.03, 0.01) |
| Sustainable General | (0.4, 0.6] | (0.5, 0.03, 0.01) |
| Less sustainable | (0.2, 0.4] | (0.3, 0.03, 0.01) |
| Poor sustainability | (0, 0.2] | (0.1, 0.03, 0.01) |

**Table 3. Evaluation index weight set.**

| Indicators | Weights | Contrast intensity | Conflictual | Comprehensive information content | Weights | Dimensionality | Dimensional weights | Weighting of indicators within dimensions |
|---|---|---|---|---|---|---|---|---|
| Food Supply Sustainability and Stability and Raw Material Supply Sustainability and Stability | w11 | 0.454 | 11.9381 | 5.418 | 0.034 | Economic dimension sustainable | 0.212 | 0.161 |
| Cost-benefit ratio of mycorrhiza cultivation | w12 | 0.430 | 12.0942 | 5.205 | 0.033 | | | 0.155 |
| Economic Situation | w13 | 0.432 | 28.3147 | 12.233 | 0.077 | | | 0.364 |
| | w14 | 0.473 | 22.6402 | 10.714 | 0.068 | | | 0.319 |
| Resource saving capacity | w21 | 0.462 | 11.816 | 5.459 | 0.034 | Ecological dimension sustainable | 0.122 | 0.283 |
| Environmental Protection Capability | w22 | 0.427 | 12.1608 | 5.197 | 0.033 | | | 0.269 |
| Natural Resource Endowment | w23 | 0.428 | 20.2132 | 8.653 | 0.055 | | | 0.448 |
| Employment poverty reduction capacity | w31 | 0.455 | 25.6928 | 11.694 | 0.074 | Social Dimension Sustainability | 0.175 | 0.422 |
| Infrastructure | w32 | 0.449 | 11.9465 | 5.367 | 0.034 | | | 0.194 |
| Public Services | w33 | 0.419 | 12.3525 | 5.171 | 0.033 | | | 0.187 |
| Nutritional health value | w34 | 0.461 | 11.819 | 5.448 | 0.034 | | | 0.197 |
| Mycorrhiza brand culture building | w41 | 0.471 | 13.1848 | 6.215 | 0.039 | Cultural dimension sustainable | 0.147 | 0.267 |
| Tourism Industry Construction | w42 | 0.434 | 12.0941 | 5.249 | 0.033 | | | 0.225 |
| Customs and Culture | w43 | 0.437 | 27.0212 | 11.821 | 0.075 | | | 0.508 |
| Financial Support | w51 | 0.437 | 27.0212 | 11.821 | 0.075 | Political dimension is sustainable | 0.192 | 0.388 |
| Endogenous power | w52 | 0.427 | 17.4638 | 7.456 | 0.047 | | | 0.245 |
| Policy support efforts | w53 | 0.479 | 12.5159 | 5.933 | 0.037 | | | 0.195 |
| International promotion efforts | w54 | 0.432 | 12.3944 | 5.260 | 0.033 | | | 0.173 |
| Practitioner quality and competence | w61 | 0.446 | 12.1687 | 12.859 | 0.081 | Technical level sustainable | 0.152 | 0.534 |
| Technology promotion and practical application | w62 | 0.460 | 28.8497 | 5.562 | 0.035 | | | 0.231 |
| Technology Development Capability | w63 | 0.431 | 12.0926 | 5.640 | 0.036 | | | 0.234 |

formula (See Table 3 for details of the weighting data for specific indicators)

$$\begin{cases} w_1 = w_{11} + w_{12} + \ldots + w_{1m} \\ w_2 = w_{21} + w_{22} + \ldots + w_{2m} \\ \ldots \\ w_n = w_{n1} + w_{n2} + \ldots + w_{nm} \end{cases} \tag{10}$$

$$w'_{11} = w_{11}/w_1, w'_{12} = w_{12}/w_1, \ldots, w'_{nm} = w_{nm}/w_n \tag{11}$$

$$W = \{w_1, w_2, \ldots, w_n\} = \{w'_{11}, w'_{12}, \ldots, w'_{nm}\} \tag{12}$$

Step 4: Determine the satisfaction evaluation composite cloud

Combining the established indicator set and weight vector set data, the numerical characteristics of the cloud model derived by the cloud algorithm, the corresponding cloud

**Table 4. Evaluation of integrated cloud parameters.**

| Indicators | Indicator cloud parameters | | | Dimensionality | Indicator cloud parameters | | |
|---|---|---|---|---|---|---|---|
| | Expectations | Entropy | Hyperentropy | | Expectations | Entropy | Hyperentropy |
| Food Supply Sustainability and Stability and Raw Material Supply Sustainability and Stability | 0.4762 | 0.464 | 0 | Economic dimension sustainable | 0.539 | 0.245 | 0 |
| Cost-benefit ratio of mycorrhiza cultivation | 0.5 | 0.418 | 0.1027 | | | | |
| Economic Situation | 0.6 | 0.376 | 0.2128 | | | | |
| | 0.5208 | 0.496 | 0 | | | | |
| Resource saving capacity | 0.5145 | 0.4788 | 0 | Ecological dimension sustainable | 0.5515 | 0.3071 | 0.0422 |
| Environmental Protection Capability | 0.4938 | 0.4101 | 0.12 | | | | |
| Natural Resource Endowment | 0.6093 | 0.3818 | 0.1936 | | | | |
| Employment poverty reduction capacity | 0.537 | 0.4644 | 0 | Social Dimension Sustainability | 0.5337 | 0.1833 | 0.0515 |
| Infrastructure | 0.566 | 0.4449 | 0.0626 | | | | |
| Public Services | 0.5128 | 0.3832 | 0.1683 | | | | |
| Nutritional health value | 0.515 | 0.4769 | 0 | | | | |
| Mycorrhiza brand culture building | 0.3332 | 0.4178 | 0.2183 | Cultural dimension sustainable | 0.5102 | 0.092 | 0.0644 |
| Tourism Industry Construction | 0.4823 | 0.4252 | 0.0869 | | | | |
| Customs and Culture | 0.6158 | 0.3859 | 0.2062 | | | | |
| Financial Support | 0.6158 | 0.3859 | 0.2062 | Political dimension is sustainable | 0.5255 | 0.0862 | 0.0556 |
| Endogenous power | 0.5625 | 0.3917 | 0.17 | | | | |
| Policy support efforts | 0.375 | 0.47 | 0.091 | | | | |
| International promotion efforts | 0.4403 | 0.4089 | 0.1405 | | | | |
| Practitioner quality and competence | 0.444 | 0.4412 | 0.063 | Technical level sustainable | 0.4287 | 0.251 | 0.0596 |
| Technology promotion and practical application | 0.324 | 0.4236 | 0.1793 | | | | |
| Technology Development Capability | 0.497 | 0.4192 | 0.0998 | | | | |

parameter matrix Z of the indicator set is generated by the inverse cloud generator algorithm as follows.

$$Z = \begin{bmatrix} c_{11} \\ c_{12} \\ \cdots \\ c_{nm} \end{bmatrix} = \begin{bmatrix} Ex_{11} & En_{11} & He_{11} \\ Ex_{12} & En_{12} & He_{12} \\ \cdots & \cdots & \cdots \\ Ex_{nm} & En_{nm} & He_{nm} \end{bmatrix} \tag{13}$$

According to the obtained weight set W and the index cloud parameter matrix Z, the result cloud model of satisfaction evaluation is expressed as

$$C = W * Z \tag{14}$$

Where the operations need to be combined with fuzzy operations, the numerical characteristics of each cloud parameter are calculated as follows.

$$\begin{cases} E_x = \sum E_{xi} w_i \\ E_n = \sqrt{\sum E_{ni}^2 w_i} \\ H_e = \sum H_{ei} w_i \end{cases} \tag{15}$$

The cloud parameters of the indicator system are detailed in Table 4.

Step 5: Determine the evaluation results and compare the clouds.

The evaluation cloud is compared with the integrated cloud C. Through the overall comparison of the cloud map, different dimensions and secondary indicators with reference to the

indicators, different cloud maps are generated using the forward cloud generator, and the corresponding integrated cloud can be compared with the satisfaction from them.

## Analysis of results

As can be seen in the MATLAB visualization Figs 1–6, it can be seen that from left to right, poor sustainability cloud, poor sustainability cloud, average sustainability cloud, good sustainability cloud, good sustainability cloud and integrated sustainability cloud of mycorrhizal industry are generated in order. According to the triple-En principle and bilateral principle, there are only a few overlapping clouds, which can be ignored. The value domain of the integrated cloud of sustainable development ability of mycorrhizal industry can be calculated according to the Formulas (7), (8) and (9) in the inverse cloud generator. the spanning degree of different cloud maps is the same [21, 22].

As can be seen from the Figs 1–6, the integrated cloud of economic-level sustainability of mycorrhizal industry is located between the average cloud of sustainability and the better cloud of sustainability, which is closer to the average cloud of sustainability, indicating that the economic-level sustainability of mycorrhizal industry still has a lot of room for growth and development, and the span of the overall integrated cloud chart is large, indicating that the development of economic-level sustainability in There is a large span difference between 2017 and 2020, and its atomization dispersion is low, indicating that overall, the economic level sustainability of mycorrhizal industry is steadily increasing year by year. The integrated cloud chart of sustainability of ecological level of mycorrhizal industry and the integrated cloud chart of sustainability of social level of mycorrhizal industry are both located between the cloud of average sustainability and the cloud of better sustainability, closer to the cloud of

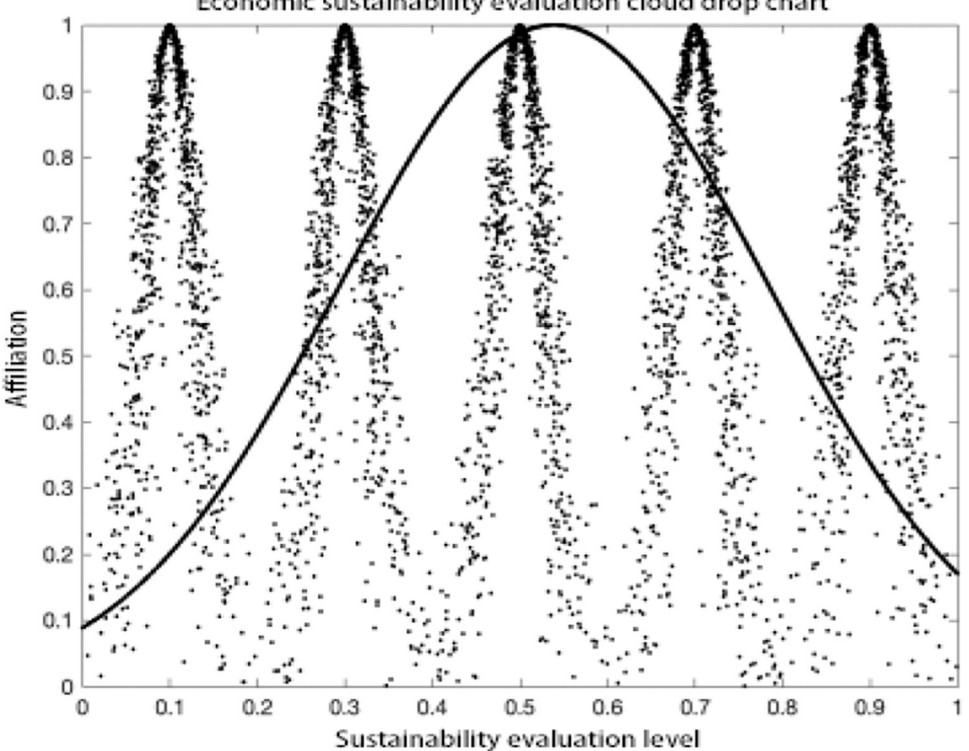

**Fig 1. Economic sustainability evaluation cloud drop chart.**

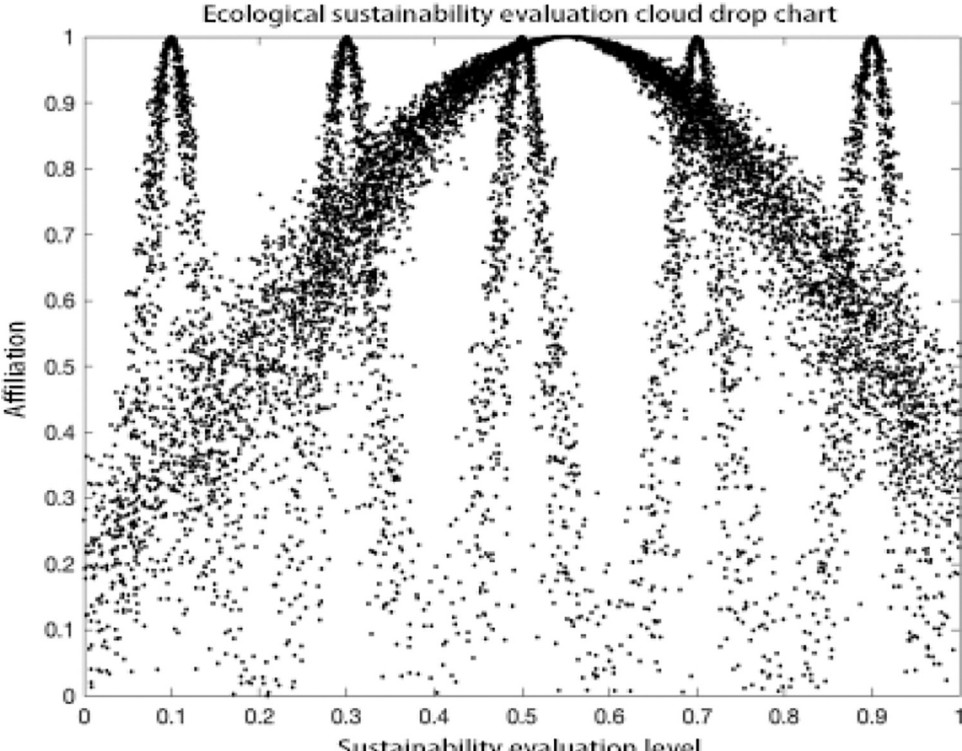

**Fig 2. Ecological sustainability evaluation cloud drop chart.**

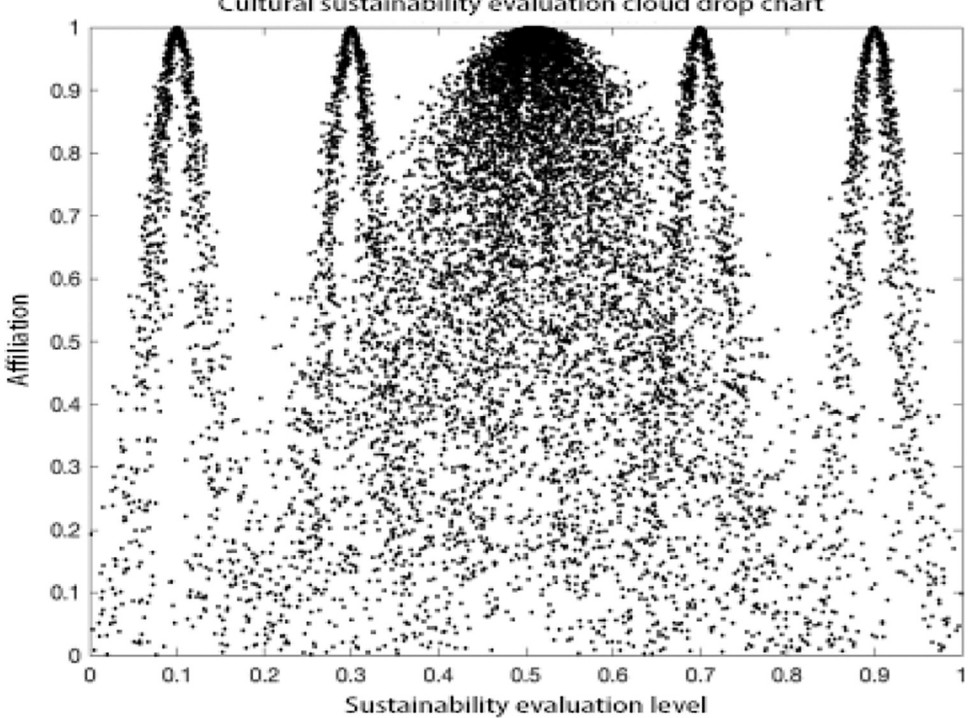

**Fig 3. Cultural sustainability evaluation cloud drop chart.**

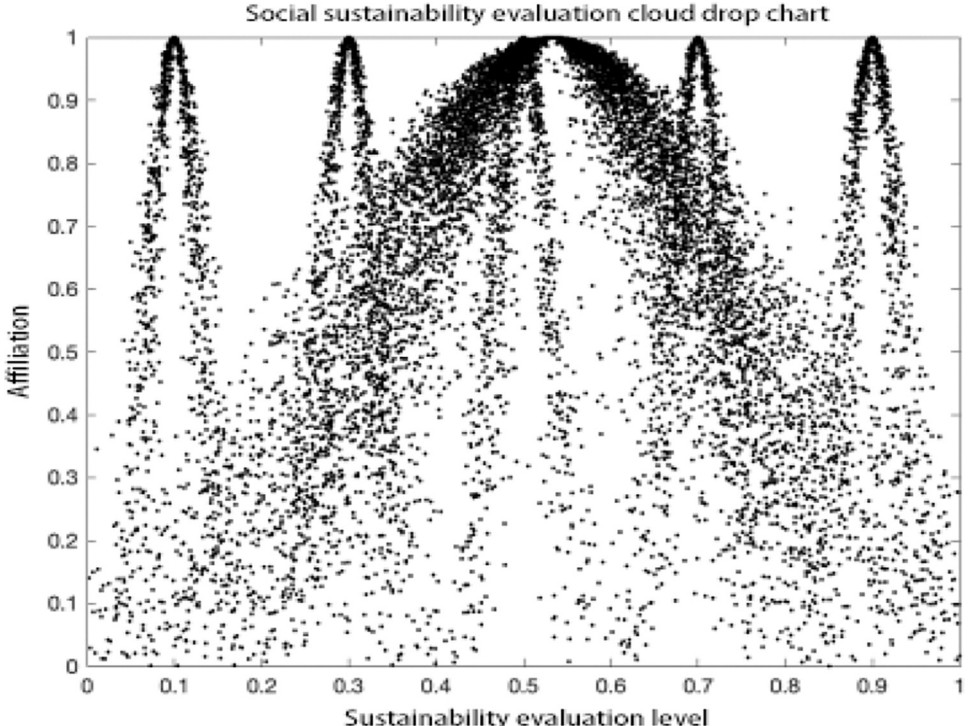

**Fig 4. Social sustainability evaluation cloud drop chart.**

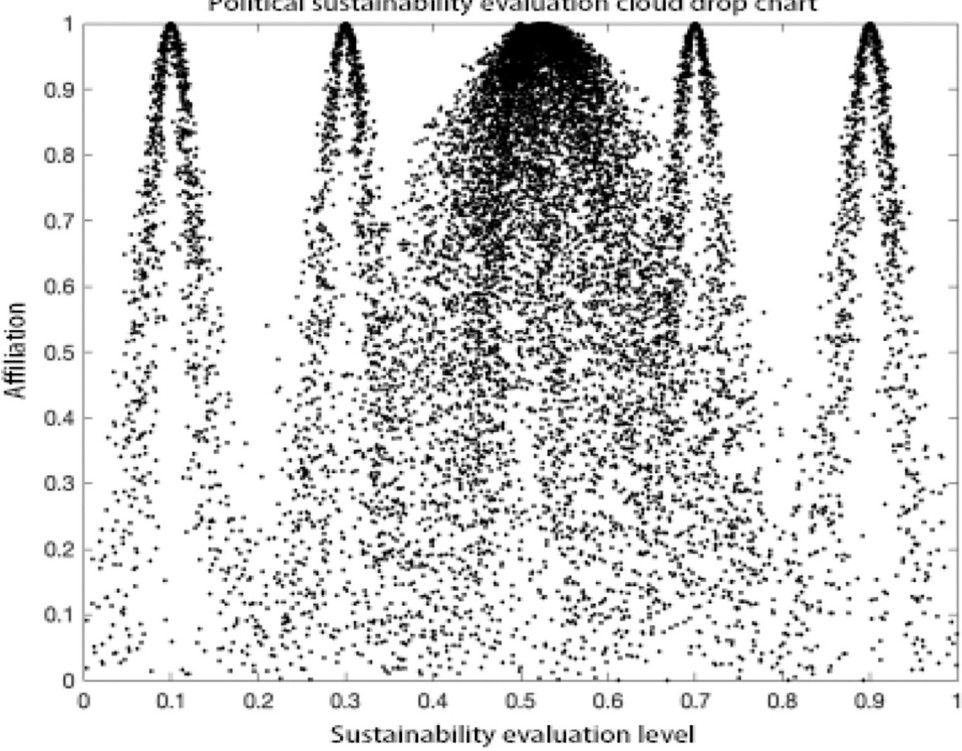

**Fig 5. Political sustainability evaluation cloud drop chart.**

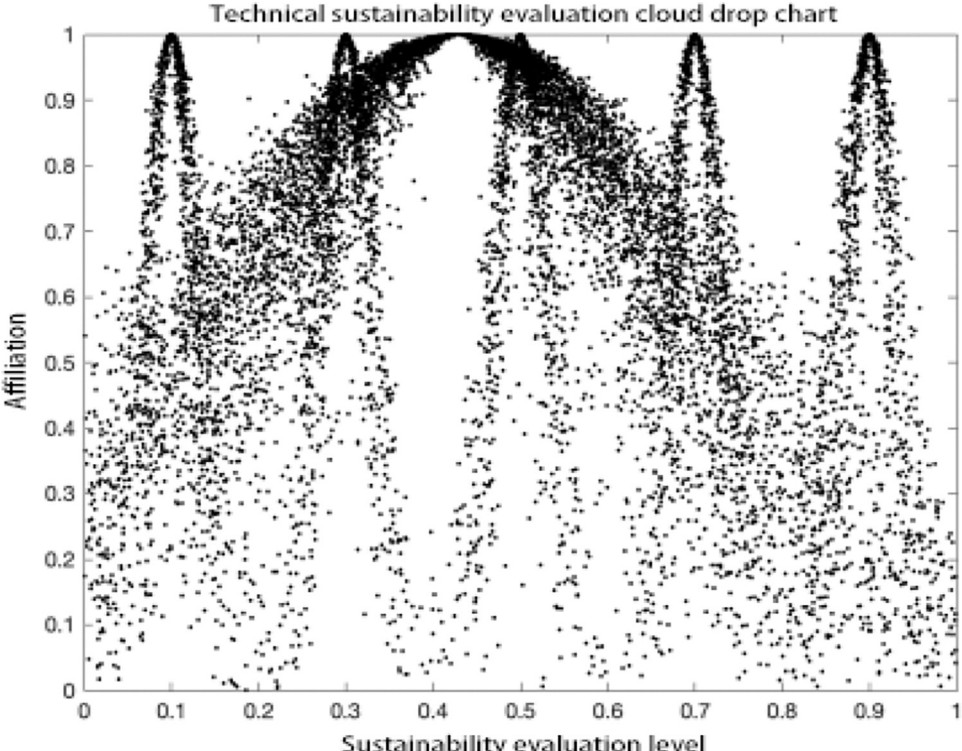

**Fig 6. Technical sustainability evaluation cloud drop chart.**

average sustainability, and both of them have a stronger degree of fogging dispersion and a large overall span as He increases, indicating that between 2017 and 2020, the The ecological level sustainability of mycorrhizal industry and the social level sustainability of mycorrhizal industry have a certain degree of span difference, which indicates that their development trend is positive, but their overall fogging dispersion is strong, which also indicates that the development status between the influencing elements of ecological level sustainability of mycorrhizal industry and social level sustainability of mycorrhizal industry is uneven. The integrated cloud diagram of cultural-level sustainability and the integrated cloud diagram of political-level sustainability of the mycorrhiza industry both lie between the average sustainability cloud and the better sustainability cloud, and are closer to the average sustainability cloud, with a strong degree of fogging dispersion but not a large overall span, indicating that the cultural-level sustainability and the political-level sustainability of the mycorrhiza industry do not differ between 2017 and There is not much difference in the development between 2020 and there is a situation that the degree of development difference between the influencing elements of cultural level and political level sustainability of mycorrhizal industry is large. The technical level sustainability integrated cloud of mycorrhizal industry is located between the poor sustainability cloud and the average sustainability cloud, closer to the average sustainability cloud, and there is a strong degree of dispersion and a large overall span, indicating that the technical level sustainability of mycorrhizal industry is poorer than the other levels of sustainability, and there is a large span of development from 2017 to 2020. The development span is large and the degree of development differences between its influencing elements is large.

## Discussion

First of all, compared with the traditional evaluation methods used in evaluation articles, such as AHP method and fuzzy mathematical method, etc. The number of cycles of the cloud model is taken as 8000, and the evaluation results obtained after thousands of mean and standard deviation operations are more reliable [21], meanwhile, the weighting method of CRITIC is added to be able to measure the weight index of each index system, so that the data can reflect the overall situation more objectively and scientifically, and make the data more representative as a whole. Secondly, in constructing the index system of sustainable development ability of mycorrhizal industry, there are not many articles of mycorrhizal industry evaluation that can be directly referred to, and there is a lack of relevant data of mycorrhizal industry, so we can only refer to some industrial evaluation articles and agroforestry development evaluation articles [23–26], and some data that are difficult to obtain directly are indirectly measured by using relevant indicators. Moreover, in order to highlight the characteristics of mycorrhizal industry evaluation system. In order to highlight the characteristics of the evaluation system of mycorrhiza industry, the functionalities of mycorrhiza itself are considered to build the characteristic indicators of sustainable development of mycorrhiza industry [2]. Overall, it is an attempt and an innovation to build the evaluation system of sustainability of mycorrhizal industry. Finally, in order to better ensure the scientific and rational construction of the index system, the policy documents related to the development of mycorrhizal industry are summarized and sorted out: from the "Grass Planting Edible Mushroom" project plan issued by Fujian Province in 1991, to the "Notice on the Pilot Work of Mycorrhizal Industry Development" in 2009, to the "Notice on the Pilot Work of Mycorrhizal Industry Development" in 2022. The changes in the content of policy documents, such as the "Fourteenth Five-Year Plan for Promoting Agricultural and Rural Modernization in Fujian Province" in 2022, all reflect that the role of mycorrhizal industry in different dimensions is getting more and more attention from Fujian Province. The general evolution process is: from the earliest emphasis on the development of mycorrhizal technology to the emphasis on the technical, economic and social aspects of the mycorrhizal industry as a whole, and finally to the emphasis on the development of mycorrhizal industry in technical, economic, social, political and cultural dimensions. The contents of these policy documents also provide important support for the evaluation system of the sustainability of mycorrhizal industry and ensure the reliability and scientificity of the selected dimensions of the system.

## Conclusion and suggestions for countermeasures

### Conclusion

The results of the above analysis show that the sustainability of the economic level of the mycorrhizal industry is developing to an average degree, but the overall development trend is good and there are not too many problems. The sustainability of ecological, social, and technological levels all have large differences in the development of indicators and the overall development status is average, but on the whole, the ecological, social, and technological levels show a steady forward development from 2017 to 2020. The cultural and political dimensions of sustainability not only have large differences in the development of indicators and an average overall development status, but also have a small development span from 2017 to 2020 and a slow overall development.

### Suggestions for countermeasures

Based on the above conclusions and the situation of the cloud parameters of the single indicator, the following countermeasures and suggestions are proposed for the improvement and development of the sustainability of the mycorrhiza industry.

First, the economic level, from the overall economic level of sustainability, mycorrhiza industry in the economic level of sustainability does not have too many problems, from the economic level of the three levels of indicators of the cloud parameters, mycorrhiza planting cost-benefit ratio and agricultural economic development situation on the economic level of sustainability has the greatest impact, its development of the degree of difference is large, these two indicators are also the impact of Therefore, it is necessary to accelerate the construction of mycorrhiza industrial chain, improve the overall added value of mycorrhiza and increase its profitability; secondly, it is necessary to focus on the development of agricultural economy, agricultural development is influenced by urbanization development, for which it is necessary to actively protect agricultural land and seek the balance between agriculture and commerce, urbanization and rural protection.

Second, the ecological, social and technological dimensions. In terms of the overall ecological, social and technological levels of sustainability, the main problem of the sustainability of the mycorrhizal industry in the ecological, social and technological levels is the large degree of variation in the development of the influencing factors. From the situation of the cloud parameters of the three levels of indicators under the ecological level, the two factors of environmental protection capacity and natural resource endowment have a large degree of development differences, which have the greatest impact on the overall ecological level of sustainability. Therefore, firstly, the ability of mycorrhiza for environmental protection should be improved, which is related to the development status of the technical level; secondly, the available area of natural resources should be improved, and the natural resources of forest land and cultivated land should be protected. From the situation of the cloud parameters of the three levels of indicators under the social level, the development of two elements, infrastructure and public services, has a greater degree of difference and has the greatest impact on the overall social level of sustainable development capacity. Therefore, it is necessary to strengthen the construction of infrastructure and public services, and provide the degree of coverage and perfection of infrastructure and public services. From the situation of the cloud parameters of the three levels of indicators under the technical level, the development of the element of technology promotion and practical application has the greatest degree of variation and has the greatest impact on the overall technical level of sustainability, and the development of the two elements of the quality and ability of practitioners and technology research and development capacity has less impact on the overall technical level of sustainability. In this regard, to strengthen the process of mycorrhizal industry technology input practice application and the promotion of mycorrhizal technology, from the actual situation of mycorrhizal industry development in Fujian Province, the current difficulties in the development of mycorrhizal industry in Fujian Province is also the bottleneck of mycorrhizal technology input actual production process.

Third, the cultural dimension and the political dimension. From the overall cultural level and political level sustainability, the main problem of mycorrhiza industry in cultural level and political level sustainability is that its development span is not large and slow. From the situation of the cloud parameters of the three levels of indicators under the cultural level, the three elements of mycorrhiza brand culture construction, custom culture and tourism industry construction have an impact on the overall slow development, but the overall trend of tourism industry development is more stable, and the difference between mycorrhiza brand culture construction and custom culture development is larger. In this regard, we should strengthen the cultural construction of mycorrhizal brand, strengthen the publicity and promotion efforts. From the situation of the cloud parameters of the three levels of indicators under the political level, the financial support, endogenous power, international promotion and policy support have an impact on the overall slow development, but the overall increase in policy support is more stable, and the development of financial support, endogenous power and

international promotion is more different, therefore, to strengthen the financial support of mycorrhiza industry, improve the leaders for mycorrhiza At the same time, although the overall increase of policy support is stable, but overall, there are not many policy documents about mycorrhizal industry in Fujian Province, and lack of target, it is necessary to make detailed sorting for the development of mycorrhizal industry, and put forward policy guidance documents with more targeted development opinions.

## Future scope and limitations of the model

As an evaluation method, the cloud model can be used alone or in combination with the assignment method, especially in DEA measurement related studies, which often calculate efficiency values after constructing a set of input-output indicators, and the measured efficiency values are often simply analyzed using charts and graphs, while the cloud model is able to fill the missing evaluation part in DEA related studies, and can conduct more in-depth data The cloud model is able to fill the missing evaluation part of DEA related research, and can analyze, compare and explore the data in more depth. It can enrich the research content of DEA measurement articles, so that the research content can be expanded into two major parts: measurement and evaluation. Cloud model as an evaluation method is also applicable to evaluate various scientific problems. Its application scope is relatively wide. However, the cloud model itself belongs to the research methods of engineering disciplines, and the use of cloud model in the social science field is still relatively scarce, and the application of cloud model in the social science field still has a lot of room for exploration. And the cloud model is more suitable for the comparison between two indicators, that is, through the evaluation of the cloud model to solve the problem of which indicator results are better. Its main function is to realize the mutual transformation of qualitative and quantitative, and it is more suitable for the evaluation and comparison of multi-dimensional and multi-indicators, while the evaluation of single indicator and single system may be contrary to the original purpose of the cloud model.

## Author Contributions

**Conceptualization:** Xue-Yuan Li, Zu-Mei Cai.

**Funding acquisition:** Sen-wei Huang.

**Resources:** Yue-Wei Hong.

**Supervision:** Sen-wei Huang.

**Visualization:** Yue-Wei Hong.

**Writing – original draft:** Qian Lin.

**Writing – review & editing:** Qian Lin.

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
