## [Decision Letter · Decision Letter 0]

5 Dec 2022

PONE-D-22-29299Evaluation of Sustainability of Mycorrhizal Industry Development in Fujian Province-Based on a Combination of CRITIC Empowerment Method and Cloud ModelPLOS ONE

Dear Dr. li,

Thank you for submitting your manuscript to PLOS ONE. After careful consideration, we feel that it has merit but does not fully meet PLOS ONE’s publication criteria as it currently stands. Therefore, we invite you to submit a revised version of the manuscript that addresses the points raised during the review process.

We look forward to receiving your revised manuscript.

Kind regards,

Dragan Pamucar

Academic Editor

PLOS ONE

Journal Requirements:

2. Our internal editors have looked over your manuscript and determined that it is within the scope of our Sustainability and the Circular Economy Call for Papers. The Collection will encompass a diverse and interdisciplinary set of submissions related to sustainability and the circular economy, focusing on production models, business plans, and the contribution of global initiatives to increased sustainability in economic, environmental, and social terms. Additional information can be found on our announcement page: Sustainability and the Circular Economy - PLOS Collections . If you would like your manuscript to be considered for this collection, please let us know in your cover letter and we will ensure that your paper is treated as if you were responding to this call. If you would prefer to remove your manuscript from collection consideration, please specify this in the cover letter.

3. In order to meet journal requirements for reporting and reproducibility, at this time we request that you please update the Methods section to report in full the original source(s) of the data and the methods used to collect it in sufficient detail for another researcher to access the same data. Please ensure that you include a statement specifying whether the collection and analysis method complied with the terms and conditions of the data source.

7. Please amend the manuscript submission data (via Edit Submission) to include author Sen-wei Huang, Zu-Mei Cai1, Yue-Wei Hong.

8. Please include a separate caption for each figure in your manuscript.

Reviewers' comments:

Reviewer's Responses to Questions

**Comments to the Author**

1. Is the manuscript technically sound, and do the data support the conclusions?

Reviewer #1: Yes

Reviewer #2: Yes

2. Has the statistical analysis been performed appropriately and rigorously? 

Reviewer #1: N/A

Reviewer #2: Yes

3. Have the authors made all data underlying the findings in their manuscript fully available?

Reviewer #1: Yes

Reviewer #2: Yes

4. Is the manuscript presented in an intelligible fashion and written in standard English?

Reviewer #1: Yes

Reviewer #2: Yes

5. Review Comments to the Author

Reviewer #1: The paper intends to investigate application of CRITIC method and cloud model for evaluation of the sustainability of mycorrhizal industry development. Presented methodology has potential in decision making processes. Hence, investigation of this topic is certainly of interest to the research community. The strengths of this paper are: Relevant topic; Flow of the paper; and Explanation of the methods. However, the author(s) need(s) to consider the following points as limitation or further scope for refining the paper:

- I suggest to the authors to clearly summarize what specific advantages the approach brings.

- Some old references can easily be removed.

- Literature review should be presented in a better way. You should discuss application of various objective and subjective methods in MCDM and supplier selection problem. You should update your literature review with a papers published in last two-three years, and remove some older references. Some new references with application of CRITIC method are missing. I suggest authors to discuss the following papers in the revised manuscript:

Mukhametzyanov, I. (2021). Specific character of objective methods for determining weights of criteria in MCDM problems: Entropy, CRITIC and SD. Decision Making: Applications in Management and Engineering, 4(2), 76-105.;

- Add future scope and limitations of the model.

Reviewer #2: Dear editor,

Thank you for sending me for review the paper “Evaluation of Sustainability of Mycorrhizal Industry Development in Fujian Province-Based on a Combination of CRITIC Empowerment Method and Cloud Model”. This paper intends to investigate extension of CRITIC method. Presented methodology has potential in decision making process and we can give support to the authors for investigation this topic. However, the author(s) need to consider the following points as limitation or further scope for refining the paper:

- Introduction should be clearly stated research questions and targets first. Then answer several questions: Why is the topic important (or why do you study on it)? What are the research questions? What are your contributions? Why is to propose this particular methods? The last two questions are answered in some parts in the Introduction section. But, the answer is not presented in a proper way. You should provide more information in this regard.

- Need to highlight the novelty of study in the introduction.

- I suggest authors to clearly summarize what specific advantages brings your approach. Enrich your Introduction section with more explanation: Why do you present this approach? Why you use CRITIC method for criteria weighting and not the other objective methods like Entropy, FANMA or LOPCOW methods?

- Why not implementation of subjective methods? Discuss more subjective methods for determining criteria weights like FUCOM, BWM, LBWA and DIBR methods.

- I must stress that there are numerous limitations of CRITIC method that authors should have on their minds. For example, if only one value in decision matrix is above/below the other values for 20-30% (within the same criteria) leads drastically to increasing criteria weight of that criteria. That limitation is presented in almost all objective methods. In my opinion this facts should not be neglected. So, why you have used objective methodology for determining weights?

- Remove lumped references. All references cited in the text should be explained and discussed in the text. Remove some old references published before 2017-2018.

- Literature review should be presented in a better way. You should discuss application of various objective and subjective methods in MCDM and supplier selection problem. You should update your literature review with a papers published in last two-three years, and remove old references. I suggest authros to discuss the following papers in the revised manuscript: Stojanović, I., & Puška, A. (2021). Logistics Performances of Gulf Cooperation Council’s Countries in Global Supply Chains. Decision Making: Applications in Management and Engineering, 4(1), 174-193.; Mukhametzyanov, I. (2021). Specific character of objective methods for determining weights of criteria in MCDM problems: Entropy, CRITIC and SD. Decision Making: Applications in Management and Engineering, 4(2), 76-105.; Badi, I., L. J. Muhammad, Mansir Abubakar, & Mahmut Bakır. (2022). Measuring Sustainability Performance Indicators Using FUCOM-MARCOS Methods. Operational Research in Engineering Sciences: Theory and Applications, 5(2), 99-116.; Alosta, A., Elmansuri, O., & Badi, I. (2021). Resolving a location selection problem by means of an integrated AHP-RAFSI approach. Reports in Mechanical Engineering, 2(1), 135-142.; Pamucar, D., Žižović, M., & Đuričić, D. (2022). Modification of the CRITIC method using fuzzy rough numbers. Decision Making: Applications in Management and Engineering, 5(2), 362-371. https://doi.org/10.31181/dmame0316102022p

- Add more deep calculations in case study section.

- The problem on which this present method is applied has significant social and managerial implications. How the method can address those implications need to be included.

- Conclusion- Add future scope. Also, how the proposed method can be applicable to other real life problems need to be mentioned. Add limitations of proposed model. Do not use bullets or numerations in this section.

I will review revised paper with my pleasure.

6. PLOS authors have the option to publish the peer review history of their article (what does this mean?). If published, this will include your full peer review and any attached files.

Reviewer #1: No

Reviewer #2: No

---

## [Author Response · Author response to Decision Letter 0]

26 Dec 2022

In response to the editor's request, the authors responded as follows.

1. The format of the paper has been revised and adjusted.

2. After discussion with the co-authors, it was decided not to publish the paper in the Sustainable Development and Circular Economy collection, and this has been added in the cover letter.

3. The data used in the article and the corresponding calculation methods and sources are available in Table 1 of the article, and have been added at the bottom of Table 1 of the article.

4. The author information has been supplemented.

5.The rest of the requirements have also been revised, if there is still that part is not done, I hope the editor can give the opportunity to revise, thank you very much!

Response to the suggestion of the first reviewer.

1. The strengths and weaknesses of the method and the scope of application have been added in the article. Supplemented in the conclusion future scope and limitations section.

Future scope and limitations of the model

As an evaluation method, the cloud model can be used alone or in combination with the assignment method, especially in DEA measurement related studies, which often calculate efficiency values after constructing a set of input-output indicators, and the measured efficiency values are often simply analyzed using charts and graphs, while the cloud model is able to fill the missing evaluation part in DEA related studies, and can conduct more in-depth data The cloud model is able to fill the missing evaluation part of DEA related research, and can analyze, compare and explore the data in more depth. It can enrich the research content of DEA measurement articles, so that the research content can be expanded into two major parts: measurement and evaluation. Cloud model as an evaluation method is also applicable to evaluate various scientific problems. Its application scope is relatively wide. However, the cloud model itself belongs to the research methods of engineering disciplines, and the use of cloud model in the social science field is still relatively scarce, and the application of cloud model in the social science field still has a lot of room for exploration. And the cloud model is more suitable for the comparison between two indicators, that is, through the evaluation of the cloud model to solve the problem of which indicator results are better. Its main function is to realize the mutual transformation of qualitative and quantitative, and it is more suitable for the evaluation and comparison of multi-dimensional and multi-indicators, while the evaluation of single indicator and single system may be contrary to the original purpose of the cloud model.

2. The literature review has been supplemented with emphasis on the application of the empowerment method and cloud model. And the articles recommended by the reviewers are cited.

The current academic research on multi-criteria decision making often uses the assignment method to divide the weights of multiple indicator systems in order to achieve the scientific nature of the evaluation results. The mainstream view is that the assignment method can be divided into two types: subjective assignment method and objective assignment method, and subjective assignment method, such as AHP method [23], FUCOM method [24], Delphi method, etc., which mainly relies on the information provided by experts to assign weights to indicators, and its shortcomings are obvious. This means that the decision maker influences the decision making process and the outcome is more influenced by the decision maker's subjective preferences. The objective empowerment method uses existing data to make the empowerment. In comparison, the advantage of the objective assignment method over the subjective assignment method is that it is not influenced by the decision maker's preferences and is more scientific and reliable in the representation of the evaluation results [25]. Although scholars have pointed out that the objective assignment methods such as entropy method and CRITIC assignment method have the same defects in practice, CRITIC assignment method is more systematic than entropy method because it not only considers the comparison intensity between indicators but also considers the conflict between indicators, and has the advantages of both entropy method and principal component analysis.The use of CRITIC assignment method can The use of CRITIC weighting method can reflect the evaluation results more accurately and objectively and scientifically [26]. The advantage of the cloud model is that it can simulate the overall results through multiple simulations, reflect the stability and scientific nature of the overall results, and can realize the mutual transformation of quantitative and qualitative through its own data. Compared with the common line graphs and other ways to interpret the evaluation results, the evaluation of the cloud model can be more scientific and reliable, and it can objectively analyze the good and bad evaluation results and visualize the results, so that it is easy to analyze and dig into the overall situation of the development of the evaluation results, the overall stability, the differences in development, the stability of the development differences, and many other information. In the current research, the empowerment method is often combined with the cloud model, and the common combination is mainly the entropy method and cloud model, and the combination of CRITIC empowerment method and cloud model. Based on the introduction of the empowerment method above and considering the main combination of scholars at present, the study adopts the combination model of CRITIC empowerment method and cloud model to analyze and evaluate the sustainability of mycorrhizal industry.

3. The article is less researched in mycorrhiza and mycorrhiza industry, so there are not many literature that can be referenced and cited. Therefore, some of the cited literature are published in earlier years, but they still provide great reference value for the research of mycorrhizal industry, so the authors did not delete them, but the authors listened to the advice of the reviewer and cited the relevant literature provided by the reviewer to supplement the content of the article.

Response to the second reviewer's suggestion.

The article has taken the expert's advice and the introduction has been revised as follows.

1. The literature review is supplemented, focusing on the application of the empowerment method and cloud model and the analysis and interpretation of the empowerment method by related studies. The articles recommended by the reviewers are also cited.

Future scope and limitations of the model

As an evaluation method, the cloud model can be used alone or in combination with the assignment method, especially in DEA measurement related studies, which often calculate efficiency values after constructing a set of input-output indicators, and the measured efficiency values are often simply analyzed using charts and graphs, while the cloud model is able to fill the missing evaluation part in DEA related studies, and can conduct more in-depth data The cloud model is able to fill the missing evaluation part of DEA related research, and can analyze, compare and explore the data in more depth. It can enrich the research content of DEA measurement articles, so that the research content can be expanded into two major parts: measurement and evaluation. Cloud model as an evaluation method is also applicable to evaluate various scientific problems. Its application scope is relatively wide. However, the cloud model itself belongs to the research methods of engineering disciplines, and the use of cloud model in the social science field is still relatively scarce, and the application of cloud model in the social science field still has a lot of room for exploration. And the cloud model is more suitable for the comparison between two indicators, that is, through the evaluation of the cloud model to solve the problem of which indicator results are better. Its main function is to realize the mutual transformation of qualitative and quantitative, and it is more suitable for the evaluation and comparison of multi-dimensional and multi-indicators, while the evaluation of single indicator and single system may be contrary to the original purpose of the cloud model.

2. The conclusion section is supplemented with the future scope and limitation section, focusing on the advantages and disadvantages of the modeling approach.

The current academic research on multi-criteria decision making often uses the assignment method to divide the weights of multiple indicator systems in order to achieve the scientific nature of the evaluation results. The mainstream view is that the assignment method can be divided into two types: subjective assignment method and objective assignment method, and subjective assignment method, such as AHP method [23], FUCOM method [24], Delphi method, etc., which mainly relies on the information provided by experts to assign weights to indicators, and its shortcomings are obvious. This means that the decision maker influences the decision making process and the outcome is more influenced by the decision maker's subjective preferences. The objective empowerment method uses existing data to make the empowerment. In comparison, the advantage of the objective assignment method over the subjective assignment method is that it is not influenced by the decision maker's preferences and is more scientific and reliable in the representation of the evaluation results [25]. Although scholars have pointed out that the objective assignment methods such as entropy method and CRITIC assignment method have the same defects in practice, CRITIC assignment method is more systematic than entropy method because it not only considers the comparison intensity between indicators but also considers the conflict between indicators, and has the advantages of both entropy method and principal component analysis.The use of CRITIC assignment method can The use of CRITIC weighting method can reflect the evaluation results more accurately and objectively and scientifically [26]. The advantage of the cloud model is that it can simulate the overall results through multiple simulations, reflect the stability and scientific nature of the overall results, and can realize the mutual transformation of quantitative and qualitative through its own data. Compared with the common line graphs and other ways to interpret the evaluation results, the evaluation of the cloud model can be more scientific and reliable, and it can objectively analyze the good and bad evaluation results and visualize the results, so that it is easy to analyze and dig into the overall situation of the development of the evaluation results, the overall stability, the differences in development, the stability of the development differences, and many other information. In the current research, the empowerment method is often combined with the cloud model, and the common combination is mainly the entropy method and cloud model, and the combination of CRITIC empowerment method and cloud model. Based on the introduction of the empowerment method above and considering the main combination of scholars at present, the study adopts the combination model of CRITIC empowerment method and cloud model to analyze and evaluate the sustainability of mycorrhizal industry.

3. In response to the expert's suggestion to add in-depth calculations in the case study section, I am very sorry that the authors did not understand the expert's suggestion well. If necessary, we hope that the experts can provide more detailed comments, and the authors will definitely learn from them and make changes. Also, the authors would like to clarify that the overall analysis is based on the paradigm of the study, and that the overall content that needs to be presented and reported has been reported and presented.

---

## [Decision Letter · Decision Letter 1]

30 Jan 2023

Evaluation of Sustainability of Mycorrhizal Industry Development in Fujian Province-Based on a Combination of CRITIC Empowerment Method and Cloud Model

PONE-D-22-29299R1

Dear Dr. li,

We’re pleased to inform you that your manuscript has been judged scientifically suitable for publication and will be formally accepted for publication once it meets all outstanding technical requirements.

Kind regards,

Dragan Pamucar

Academic Editor

PLOS ONE

Additional Editor Comments (optional):

Reviewers' comments:

Reviewer's Responses to Questions

**Comments to the Author**

1. If the authors have adequately addressed your comments raised in a previous round of review and you feel that this manuscript is now acceptable for publication, you may indicate that here to bypass the “Comments to the Author” section, enter your conflict of interest statement in the “Confidential to Editor” section, and submit your "Accept" recommendation.

Reviewer #1: All comments have been addressed

Reviewer #2: All comments have been addressed

2. Is the manuscript technically sound, and do the data support the conclusions?

Reviewer #1: Yes

Reviewer #2: Yes

3. Has the statistical analysis been performed appropriately and rigorously? 

Reviewer #1: N/A

Reviewer #2: Yes

4. Have the authors made all data underlying the findings in their manuscript fully available?

Reviewer #1: Yes

Reviewer #2: Yes

5. Is the manuscript presented in an intelligible fashion and written in standard English?

Reviewer #1: Yes

Reviewer #2: Yes

6. Review Comments to the Author

Reviewer #1: (No Response)

Reviewer #2: The authors have addressed the point of my concern. I am happy with their corrections. Hence, I would like to recommend this manuscript to be published.

7. PLOS authors have the option to publish the peer review history of their article (what does this mean?). If published, this will include your full peer review and any attached files.

Reviewer #1: No

Reviewer #2: No

---

## [Editor Report · Acceptance letter]

2 Feb 2023

PONE-D-22-29299R1 

Evaluation of Sustainability of Mycorrhizal Industry Development in Fujian Province-Based on a Combination of CRITIC Empowerment Method and Cloud Model 

Dear Dr. Li:

I'm pleased to inform you that your manuscript has been deemed suitable for publication in PLOS ONE. Congratulations! Your manuscript is now with our production department. 

Kind regards, 

on behalf of

Dr. Dragan Pamucar 

Academic Editor

PLOS ONE